# Genetic and stochastic influences upon tumor formation and tumor types in Li-Fraumeni mouse models

Chang S Chan[1,2] ⓘ, Yvonne Sun[3], Hua Ke[1], Yuhan Zhao[1] ⓘ, Merzu Belete[1] ⓘ, Cen Zhang[1], Zhaohui Feng[1], Arnold J Levine[4] ⓘ, Wenwei Hu[1] ⓘ

*p53* is the most frequently mutated gene in human cancers. Li-Fraumeni syndrome patients inheriting heterozygous *p53* mutations often have a much-increased risk to develop cancer(s) at early ages. Recent studies suggest that some individuals inherited *p53* mutations do not have the early onset or high frequency of cancers. These observations suggest that other genetic, environmental, immunological, epigenetic, or stochastic factors modify the penetrance of the cancerous mutant Tp53 phenotype. To test this possibility, this study explored dominant genetic modifiers of *Tp53* mutations in heterozygous mice with different genetic backgrounds. Both genetic and stochastic effects upon tumor formation were observed in these mice. The genetic background of mice carrying *Tp53* mutations has a strong influence upon the tissue type of the tumor produced and the number of tumors formed in a single mouse. The onset age of a tumor is correlated with the tissue type of that tumor, although identical tumor tissue types can occur at very different ages. These observations help to explain the great diversity of cancers in different Li-Fraumeni patients over lifetimes.

## Introduction

Mutations in the *p53* gene are the single most common spontaneous genetic alterations observed in human cancers (Olivier et al, 2002). Serous ovarian cancers have almost 100% spontaneous somatic *p53* mutations, colon cancers have about 80%, pancreatic cancers have 70%, non-small cell lung adenocarcinomas have about 70%, and triple negative breast cancers have about 60% *p53* mutations (Cancer Genome Atlas Network, 2012). Approximately one in 20,000 individuals inherit heterozygous *p53* mutations, resulting in early onset and high frequency of cancers in each patient over a lifetime (Li & Fraumeni, 1969). These individuals are first diagnosed with early-onset cancers and a family history of selected cancers; germ line *Tp53* mutations are subsequently detected with 70–90 percent penetrance for the development of a cancer, depending upon the Tp53 allele and the sex of the patient (Malkin et al, 1990; Guha & Malkin, 2017). Recent studies have suggested that inherited *p53* mutations may occur at frequencies of 1/500 to 1/2,000 individuals in the general population without the early onset or high frequency of cancers (de Andrade et al, 2017, 2019). If this is correct, these data suggest that other genetic, environmental, immunological, epigenetic, or stochastic factors modify the penetrance of the cancerous mutant Tp53 phenotype (Shi et al, 2009; Chan, 2017).

The best test of these alternatives in humans comes from a pair of Malaysian identical twins with a germ line *Tp53* mutation. Now in their early 20s, one of the twins has had three independent cancers, whereas the other has had none (Ariffin et al, 2014). This observation favors the idea that stochastic or environmental effects (even possibly due to the treatment of the first cancer) influence the penetrance of the predisposed cancers in individuals with *p53* mutations. Other studies in humans (Bond et al, 2004) and mice (Post et al, 2010) suggest that single nucleotide polymorphisms that regulate the levels of the MDM-2 E3 ubiquitin ligase, which in turn regulate p53 levels, can modify the age of onset and penetrance of cancers. Background also makes a difference as p53 heterozygous mice in the Balb/cJ background are predisposed to breast cancers (Koch et al, 2007). Studies of F-1 and F-2 hybrids indicate a dominant acting modifier on chromosome 7 and a recessive one on the X chromosome (Koch et al, 2007). The number of examples of these types of studies is limited in humans and has not been systematically tested in multiple strains of mice. The research reported here describes a more detailed analysis of these questions by exploring dominant genetic modifiers of *Tp53* mutations in heterozygous mice derived from F-1 crosses of homozygous p53 R172H C57Bl/6 male mice with p53 wild-type mice from seven different genetic backgrounds.

The phenotypes of the tumors that were produced in the F-1 mice were quantified for the time of tumor formation or detection, the histopathological confirmation of a tumor type, and survival of the mice as well as the penetrance of the tumors formed. Evidence is presented for both genetic and stochastic effects upon tumor

[1]Rutgers Cancer Institute of New Jersey, Rutgers, State University of New Jersey, New Brunswick, NJ, USA    [2]Department of Medicine, Rutgers Robert Wood Johnson Medical School, Rutgers, State University of New Jersey, New Brunswick, NJ, USA    [3]PMV Pharma, Cranbury, NJ, USA    [4]Simons Center for Systems Biology, Institute for Advanced Study, Princeton, NJ, USA

Correspondence: alevine@ias.edu; wh221@cinj.rutgers.edu; chanc3@cinj.rutgers.edu

formation in these diverse groups of mice. Sexual dimorphism is a variable in some genetic backgrounds, and the age of onset of a tumor is correlated with the tissue type of that tumor, although identical tumor tissue types can be produced at very different ages. The genetic background of mice carrying *Tp53* mutations has a strong influence upon the tissue type of the tumor produced as well as the formation of multiple tumors produced in a single mouse. These observations, in F-1 mice with *Tp53* mutations, help to explain the great diversity of cancers in different Li-Fraumeni patients over their lifetimes.

## Results

This study examined the impact of the genetic background of the F-1 Li-Fraumeni mouse model on tumor formation. To explore the contributions of diverse genetic backgrounds upon the predisposition of inheriting a single Tp53 mutant allele to tumor formation in mice, homozygous Tp53 R172H male mice in a C57Bl/6 genetic background were crossed with wild-type 129Sl, A/J, Balb/cByJ, C3H, DBA, NOD, and SWR female mice to generate F-1 hybrids heterozygous for the Tp53R172H mutation. A total of 327 F-1 mice with these different genetic mixtures were born and survived. These crosses were carried out to look for possible dominant genetic phenotypes from the wild-type mice upon a single Tp53 mutant allele. The number of F-1 hybrid mice produced from each of these crosses were 129Sl, 49; A/J, 30; Balb/cByJ, 47; C3H, 26; DBA, 28; NOD, 76; and SWR, 71. There was a sex distortion in SWR F-1 mouse strain born which survived through nursing with the final ratio of females to males at 23/48. This is likely due to the exencephalic (failure to close the cranium at birth) phenotype previously observed in female C57Bl/6 mice (Sah et al, 1995). It is of some interest that this sex-limited phenotype only expresses itself in some inbred strains.

Humans with Li-Fraumeni syndrome develop a set of 7–8 different tumor types at early ages at an excess risk of about 100-fold higher than the general population without an inherited p53 mutation (Levine, 2020). For that reason, several different phenotypes were explored in the population of F-1 mice produced in this experiment. The first was the time after birth of tumor development,

as determined by randomly euthanizing the above-described F-1 mice at different times and determining by anatomical observation if a growth of tumor was observed. Based upon previous observations (Venkatachalam et al, 2001; Donehower & Lozano, 2009), this was performed most intensively between 11 and 20 mo after birth. This allows time for a reduction to homozygosity of the mutant Tp53 allele (commonly by the loss of the wild-type Tp53-containing chromosome) and additional mutations that together give rise to a malignant tumor. Fig 1 shows the cumulative percentage of mice in each of the seven different F-1 cohorts that develop an enlarged organ (lymph node, spleen), or had a frank tumor based upon anatomical observations. Each suspected tumor was excised, placed in a fixative, embedded, sectioned, and examined by a pathologist to confirm that the tumor was a benign or malignant cancer and to classify the tissue type of the tumor.

Fig 2A shows the cumulative percentage of mice in each of the seven different cohorts as determined by histopathological determination and mirrors the data in Fig 1. A panel of representative H&E images of tumors, including thymic lymphoma, adenocarcinoma, sarcoma, osteosarcoma, liposarcoma, and angiosarcoma, are presented in Fig 2B. In Figs 1 and 2, the A/J F-1 hybrid mice produced detectable tumors at earlier times than did the NOD F-1 hybrid mice, based upon both gross anatomical and histopathological analysis (*P* < 0.007). The phenotypes deduced from Figs 1 and 2 are the earliest detection of tumor development in situ.

A more common measure of the timing of tumor development in mouse strains is the age of the percent survival of the mice, which are euthanized when they have obvious effects of their tumor and/or reach humane end point, which is then confirmed by histopathological analysis. To determine that phenotype, the Tp53R172H homozygous male mice were mated to either A/J wild type-mice or NOD wild-type mice, and the F-1 mice produced (A/J, N = 53, and NOD, N = 54) were examined for the impact of tumor development upon the timing of their survival (Fig 3). A/J F-1 and NOD F-1 hybrid mice were chosen because these two strains showed the biggest difference in the onset of tumor formation as shown in Fig 1; AJ was the fastest strain to develop tumors in p53 heterozygous mice while NOD is the slowest strain. The mean survival of the A/J F-1 mice was 12.97 mo and that of the F-1 NOD mice was 16.90 mo (*P* < 0.0001). The

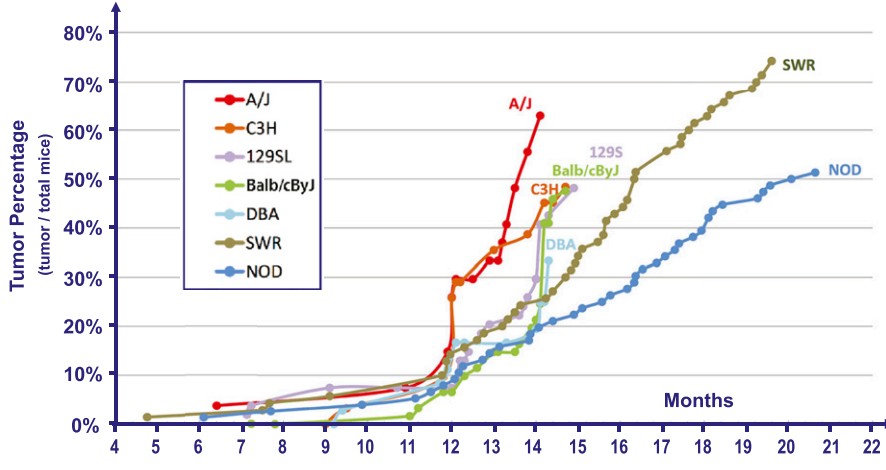

**Figure 1. Percentage of tumors in each of the F-1 strains observed by gross anatomical observation upon autopsy.**
Homozygous C57/Bl6 male mice with a R172H Tp53 mutation were crossed to seven different wild-type strains of mice as indicated in the figure. At various times after birth indicated in the time scale, some mice from each of the F-1 strains were euthanized and autopsied for the presence of gross anatomical tumors or enlarged organs in many different tissues. The graphs for each F-1 mouse strains are the cumulative percentage of the total mice in the F-1 cohort (given in the text) over a 20- to 22-mo period.

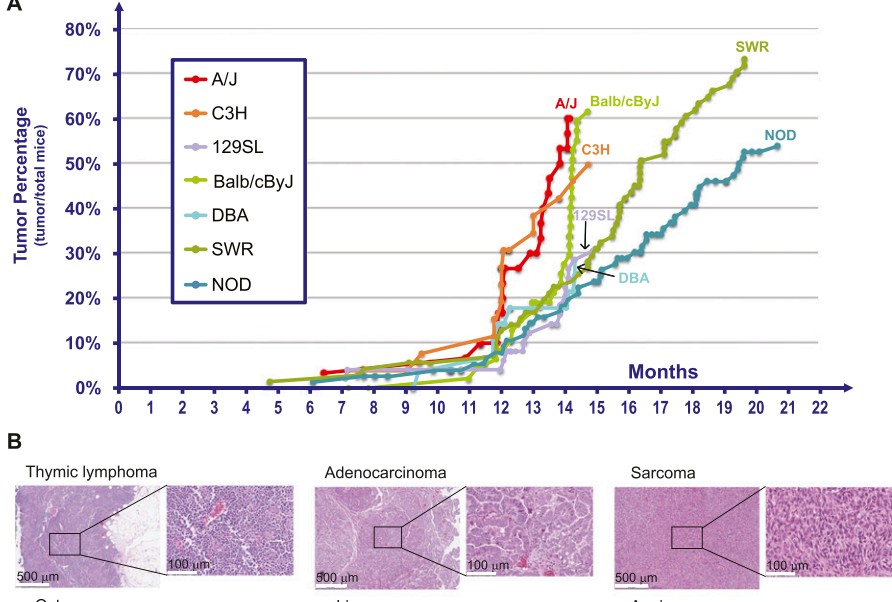

**Figure 2. Percentage of tumors in each of the different F-1 strains of mice observed by histopathological analysis of the mice indicated in Fig 1.**
The genetic crosses and the times of euthanasia of individual mice from each F-1 strains are as indicated in Fig 1. The anatomical tumors identified in Fig 1 were fixed, embedded, and sectioned and evaluated by a pathologist for both benign and malignant tumors and the tissue type. **(A)** Percentage of tumors in each of the different F-1 strains of mice. **(B)** The representative H&E images of tumors.

penetrance of the Tp53 mutant allele to produce a tumor at 20 mo after birth was for the A/J F-1 mice 87% and for the NOD F-1 mice 69%. These different phenotypes indicate that there are dominant genetic modifiers that can influence the times of the initiation and development of tumors and the survival of the mice with tumors.

Table 1 lists the number of different tissue types of tumors produced in the different F-1 mouse strains with the Tp53R172H mutant allele. The most common tumors in mice, T-cell lymphomas, are not found in humans with Li-Fraumeni syndrome but all the sarcomas are commonly found in humans (Levine, 2020). It has been observed that some of the inbred strains, such as Balb/c and C57BL/6, are predisposed to developing certain spontaneous tumors, with lymphoma often being most common, indicating that the

mouse and human T-cell adaptive immune systems have a differential dependence upon p53 to prevent a lymphoma or leukemia (Kohnken et al, 2017). Quite clearly, the distributions of tissue tumor types are influenced by the inbred strains of the F-1 cross with the Tp53 mutation from C57Bl/6 mice. Liposarcomas are over-represented in A/J F-1 mice compared with other strains ($P < 0.027$) and Balb/cByJ F-1 mice under-represent lymphomas ($P < 0.16$) and over-represent angiosarcomas ($P < 0.031$).

Upon autopsy, a number of mice had more than one tumor. This varied from a high of 15% (SWR F-1) and 11% (Balb/cByJ F-1) to a low of 0% (129Sl F-1). Fig 4 presents these data for all the F-1 hybrid strains. Fig 4 also compares the sexes of Tp53 mutant F-1 mice of different strains developing tumors by 20 mo of observation. The

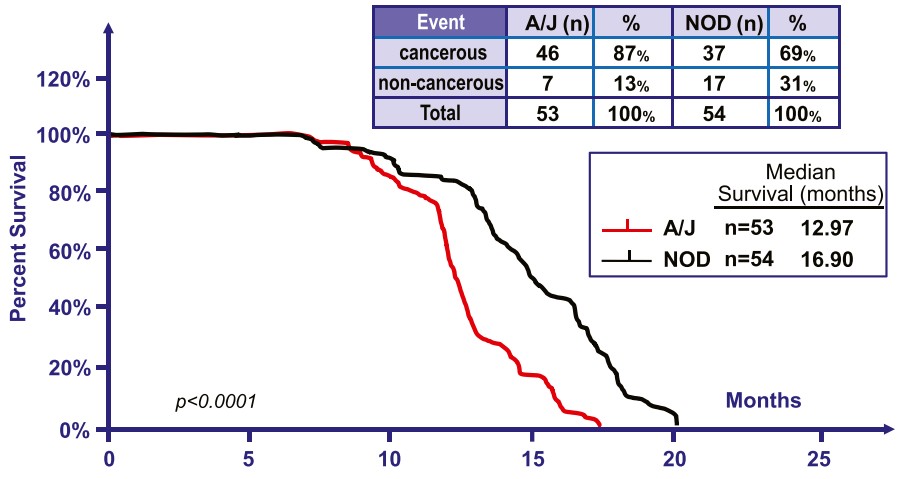

| Event | A/J (n) | % | NOD (n) | % |
|---|---|---|---|---|
| cancerous | 46 | 87% | 37 | 69% |
| non-cancerous | 7 | 13% | 17 | 31% |
| Total | 53 | 100% | 54 | 100% |

| | | Median Survival (months) |
|---|---|---|
| A/J | n=53 | 12.97 |
| NOD | n=54 | 16.90 |

p<0.0001

**Figure 3. Differences in survival in A/J and NOD F-1 mice with a heterozygous germ line mutant Tp53 gene.**
Male C57Bl/6 mice homozygous for the R172H Tp53 mutation were crossed with A/J mice and NOD mice, respectively. Life span of mice in each cohort was recorded and the histopathological analysis was performed by a pathologist. In A/J F-1 cohort, 46 mice developed cancers and in NOD F-1 cohort, 37 mice developed cancers.

**Table 1. The tumor spectrum and incidence of the different strains of F-1 mice with Tp53 mutations.**

| Hybrid Strain | Lymphoid hyperplasia/ lymphoma | Osteosarcoma | Adenocarcinoma | Spindle cell sarcoma/ sarcoma | Angioma/ angiosarcoma | Squamous carcinoma/ carcinoma | Hemangioma | Lipoma/ lipomatous tumor/ liposarcoma |
|---|---|---|---|---|---|---|---|---|
| 129SL | 10% | 6% | 2% | 0% | 2% | 2% | 2% | 2% |
| A/J | 10% | 17% | 3% | 7% | 3% | 0% | 0% | 20% |
| Balb/cByJ | 4% | 11% | 13% | 4% | 15% | 6% | 2% | 0% |
| C3H | 23% | 8% | 4% | 4% | 0% | 0% | 0% | 0% |
| DBA | 21% | 4% | 7% | 0% | 4% | 0% | 4% | 4% |
| NOD | 24% | 16% | 3% | 8% | 1% | 1% | 3% | 0% |
| SWR | 30% | 20% | 11% | 4% | 4% | 8% | 1% | 0% |
| Average incidence of tumor types | 18% | 13% | 6% | 4% | 4% | 3% | 2% | 2% |
| Total tumors (n) | 58 | 42 | 21 | 14 | 14 | 11 | 6 | 7 |

The table presents the incidence of different tumor types produced by the R172H heterozygous F-1 mouse strains. Total tumors (n) are the total number of tumors for each tumor type across all strains.

SWR F-1 mice demonstrated female lethality shortly after birth, likely due to the exencephalic phenotype. 23 females and 48 males with a Tp53 mutation survived (significant difference in sex ratio, $P < 0.0021$, Cumulative Binomial) and were observed for 20 mo. During that time, 22/23 (96%) females developed tumors, whereas 30/48 (62%) males developed tumors ($P < 0.0035$, two-tailed Fisher's exact test) (Fig 4). This was the most dramatic examples of female mice developing tumors with a higher frequency and penetrance than male mice. For Balb/cByJ F-1 hybrids with a *Tp53* mutation, 13/27 female mice and 16/20 male mice developed tumors ($P < 0.036$, two-tailed Fisher's exact test), whereas the rest of the F-1 strains did not show a pronounced sexual dimorphism for either penetrance or frequency of tumor formation. For all Tp53 mutant heterozygous strains of F-1 mice produced in this experiment, 90/149 female mice developed tumors (60%) and 98/178 male mice (55%) developed tumors over 20 mo of observation showing no significant difference ($P < 0.37$, Fisher Exact Test). The reason that the experiment was stopped at 20 mo after birth is that many of these mouse strains spontaneously develop tumors in old age (Brayton et al, 2012), which could distort the impact of an inherited *Tp53* mutation upon tumor formation. For Li-Fraumeni syndrome with an inherited *Tp53* mutation in humans, the penetrance of tumors is higher for females (93%) than males (about 70%), but this is largely due to a very high incidence of ER+, PR+ breast cancer in females (50–70% of all females with a Tp53 mutation) at young ages (20–45 yr old), whereas at older ages the frequency of cancers falls and resembles that of the general population (Levine, 2020).

Although these results are consistent with a series of genetic modifiers that influence the age of onset of a tumor, the tumor tissue type and even the sex of the host, the results of this experiment also support stochastic events in the frequencies, penetrance, and tissue types of tumors produced. The most obvious stochastic event is a spontaneous mutation in one of many different tissue specific stem cells of the body. However, there are many environmental effects such as radiation, mutagen exposure,

or the microbiome that may differ from one cage of mice to another or among inbred strains of mice, the epigenetic program, which differs between sexes and can differ among inbred strains, or by stochastic errors in development, and the adaptive immune system whose B and T-cell receptors can differ even between identical strains of mice or monozygotic twins. All of these variables can contribute to the observation that within a common pool of F-1 mice from the same strain, an identical tumor type can arise as early as 5–10 mo of age in one mouse and 15–20 mo of age in another almost genetically identical mouse. Some examples of age dependent impacts have been observed. For example, the lymphomas that form in these inbred strains tend to be predominantly T-cell thymic lymphomas at young ages and the lymphomas that are produced at older ages are mainly B-cell lymphomas. However, many of the sarcomas produced in these mice and the adenocarcinomas and squamous carcinomas observed at young or at older ages are histopathologically identical tumors. The same tumor has been detected both at a young age and late in life. Thus, both genetic and stochastic influences contribute to the production of tumors with time and tissue type in Tp53 mutant Li-Fraumeni mice.

## Discussion

In humans with Li-Fraumeni syndrome caused by an inherited *Tp53* mutation, the *Tp53* gene deficit can result in many diverse tumor types. Eight tumor types are most commonly observed, with a very high excess risk of 100–250-fold above the general population (Levine, 2020). These eight tumors are adrenal cortical carcinomas (6 mo to 4 yr, at 200-fold excess risk), choroid plexus carcinomas (6 mo to 3 yr, at 200-fold excess risk), medullary blastomas (2–5 yr, at 200-fold excess risk), rhabdomyosarcomas (1–5 yr, at 200-fold excess risk), and osteogenic sarcomas (5–20 yr, at 200-fold excess risk), ER+, PR+ breast cancers in up to 70% of females with an inherited *Tp53* mutation

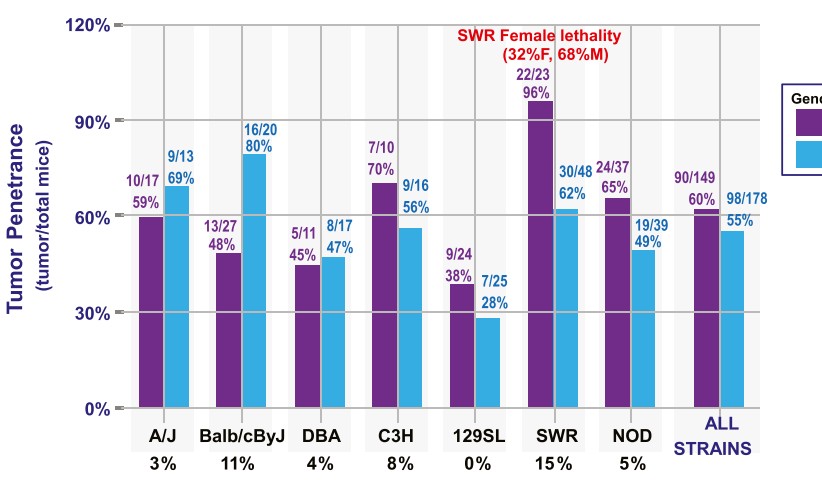

**Figure 4.  Tumor penetrance, sexual dimorphism, and multiple tumors in the different F-1 mice from the crosses in Fig 1.**

between 18 and 45 yr, lipomas and liposarcomas (20–50 yr, with an excess risk of 100-fold), leiomyosarcomas (20–60 yr, at excess risk of 100-fold), and at 50–75 yr of age, a variety of tumors from endoderm derived tissue types (colon, lung, pancreatic adenocarcinomas with a two to fourfold excess risk of cancers). With the exception of early onset thymic lymphomas in the mice, many of these tumor types are observed in the mouse models that are explored here.

C57Bl/6 mice with an inherited homozygous loss of both *Tp53* genes develop CD-4, CD-8 double positive thymic T-cell lymphomas within several weeks after birth as detected by a clonal expansion of T-cell receptor sequences enriched in the population of T-cells in the thymus. Multiple independent cancerous events occur in different T-cells (T-cells with different T-cell receptors) and these different T-cell clones then compete giving rise to a dominant clonal expansion observed by several months after birth (Dudgeon et al, 2014). These tumors progress through a series of mutations that are selected in a specific order. The *Tp53* mutation is inherited, and therefore, is an initial truncal mutation. This is followed early in development of the mouse with deletions and point mutations in the *Phosphatase and Tensin Homolog* (*PTEN*) gene that are identical in multiple malignant T-cell clones in the same mouse, but differ between different mice. This suggests that these mutations occur early in T-cell stem cell development so that different cancerous clones all develop from a single clone. After birth, these mutations are followed by the selection for gene amplifications in *cyclin D1, 2, or 3* genes and *CDK-6* genes. This happens after the *Tp53* and *PTEN* mutations because the *cyclin D1, D2*, and *D3* gene amplifications differ in different T-cell clones of the thymic lymphomas. Finally, Notch pathway mutations are detected in the T-cell clones that are selected for and dominate the tumor population (Dudgeon et al, 2014). The selection of mutations occurs in a preferred order, even though the mutations themselves can occur in a random fashion. The selection of specific mutations in a defined order is now a common observation with a number of tumor types (Levine et al, 2019; Levine, 2020). The order of mutations arises from the selection of each mutation in a tissue specific stem or progenitor cell and the recognition that this selection is specific to both the mutant gene and tissue type (Haigis et al, 2019).

This is the reason why the same mutation in the inherited form, when it acts as an initial or truncal mutation in a tissue specific stem cell, causes a different tumor than when it arises as a spontaneous mutation in a later order of selected mutations (retinoblastomas, *Tp53* mutations, *PTEN* mutations, *BRCA-1, 2* mutations, etc.). For example, inherited *p53* mutations act as truncal mutations in the list of high-risk early onset tumors described above. But these same mutations are late selected mutations in endoderm derived tumors, which occur at lower excess risks and at older ages (colon, pancreatic, lung) where the *p53* mutation brings about the malignant transformation event late in tumor development (Levine, 2020).

The approach described in this article can lead to the identification of the gene or genes that predispose to early onset inherited tumors, the selection of the tissue type of a tumor, and enhancement of the penetrance of a tumor. Dominant mutations are more readily mapped and identified than recessive ones, and these genetic crosses observe only dominant phenotypes. The sequencing of the tumors produced helps to identify oncogenes and tumor suppressor genes whose mutations act with *Tp53* mutations to give rise to a benign and malignant tumor. The driver mutations in identical tumor tissue types may indicate a preferred order of selection or a completely stochastic and random order giving rise to a tumor at a later time or not. In all these cases knowing what phenotype to explore in these genetic crosses can illuminate different aspects of tumorigenesis. The inherited mutations identify the tissue specific stem cells and tissue specific tumors where initial or truncal mutations are selected for and function by producing an expansion of benign clones of cells that form the beginning of an evolutionary process of tumor formation (Weissman, 2015; Levine et al, 2019; Levine, 2020).

## Materials and Methods

### Mouse strains and survival monitoring

Tp53[R172H/R172H] mice were gifts from Dr Gigi Lozano at MD Anderson Cancer Center (Lang et al, 2004). Mouse strains with different genetic background, including A/J (Stock No. 000646, RRID:IMSR_JAX:000646), NOD/ShiLtJ (Stock No. 001976, RRID:IMSR_JAX:001976), BALB/cByJ (Stock No. 001026, RRID:IMSR_JAX:001026), 129S1/SvlmJ (Stock No. 002448,

RRID:IMSR_JAX:002448), C3H/HeJ (Stock No. 000659, RRID:IMSR_JAX:000659), DBA/1J (Stock No. 000670, RRID:IMSR_JAX:000670), and SWR/J (Stock No. 000689, RRID:IMSR_JAX:000689), were purchased from The Jackson Laboratory. The F-1 mice with germ line Tp53$^{R172H}$ in different genetic backgrounds were generated by crossing Tp53$^{R172H/R172H}$ with mice described above in different genetic background. The health conditions of mice were checked at least twice a week and the mice were euthanized at different times or when mice reached the humane end point. The lifespans of mice were recorded as previously described (Zhao et al, 2018). All animal experiments were approved by the institutional animal care and use committee of Rutgers University.

### Autopsy and histopathological analysis

The mouse autopsy was performed as previously described (Fiette & Slaoui, 2011). In brief, the postmortem mouse was examined according to the following major steps: opening of the abdominal cavity, thoracic cavity, skull, examination of muscles, and skeleton. The major organs (liver, kidneys, and spleen) were weighed. Any tissue or organ with grossly noticeable changes were recorded and collected for histopathological evaluation. Tissue samples were fixed in formalin (10%, buffered) for 24–48 h and processed to make formalin-fixed and paraffin-embedded tissue sections. H&E staining and whole slide image scanning were performed by Histowiz company (www.histowiz.com). The scanned images were evaluated by a board-certified pathologist.

### Statistical analysis

The data were present as mean ± SD. The lifespan of mice were summarized by Kaplan–Meier plots and compared using the log-rank test using GraphPad Prism software. Statistical significance of female to male sex ratio of mice born deviating from the null model of 1:1 is computed using the cumulative binomial distribution. Statistical significance of differences in tumor penetrance between male and female mice were computed using the two-tailed Exact Fisher Test. Statistical test for increased or decreased tumor type incidence in a mouse hybrid strain was computed assuming a Gaussian distribution for tumor incidence across all strains. Values of $P < 0.05$ were considered to be significant.

## Supplementary Information

## Acknowledgements

The authors would like to thank Dr Adel El-Naggar, Professor of Pathology at MD Anderson Cancer Center, for all of his efforts in reading and interpreting the slides of mouse tissues carried out in this study. This was an enormous effort carried out with the highest quality and cooperative spirit. The authors thank Suzanne Christen for her help in preparation of the manuscript. This research was supported by program project grant from the National Institutes of Health, National Cancer Institute (P01CA087497-18, R01CA203965, and DoD W81XWH-18-10238) CS Chan would like to thank the support of Rutgers Cancer Institute and the Robert Wood Johnson Foundation.

## Author Contributions

CS Chan: conceptualization, formal analysis, and writing—original draft.
Y Sun: data curation and investigation.
H Ke: formal analysis.
Y Zhao: data curation and investigation.
M Belete: formal analysis.
C Zhang: data curation and investigation.
Z Feng: data curation, investigation, and writing—review and editing.
AJ Levine: conceptualization, funding acquisition, investigation, and writing—original draft.
W Hu: conceptualization, data curation, funding acquisition, investigation, and writing—original draft.

## Conflict of Interest Statement

The authors declare that they have no conflict of interest.

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
