## [Reviewer comments · Life Science Alliance]

Life Science Alliance

Genetic and Stochastic Influences upon Tumor Formation and Tumor Types in Li-Fraumeni Mouse Models

Chang Chan, Yvonne Sun, Hua Ke, Yuhan Zhao, Merzu Belete, Cen Zhang, Zhaohui Feng, Arnold Levine, and Wenwei Hu

DOI: <https://doi.org/10.26508/lsa.202000952>

Corresponding author(s): Wenwei Hu, Rutgers Cancer Institute of New Jersey; Arnold Levine, Institute for Advanced Study and Chang Chan, Rutgers Cancer Institute of New Jersey

Review Timeline:	Submission Date:	2020-11-03
	Editorial Decision:	2020-12-08
	Revision Received:	2020-12-14
	Accepted:	2020-12-15

Scientific Editor: Shachi Bhatt

Transaction Report:

December 8, 2020

RE: Life Science Alliance Manuscript #LSA-2020-00952-T

Dr. Wenwei Hu
Rutgers Cancer Institute of New Jersey
Radiation Oncology
195 Little Albany Street
New Brunswick 8903

Dear Dr. Hu,

Thank you for submitting your revised manuscript entitled "Genetic and Stochastic Influences upon Tumor Formation and Tumor Types in p53 inherited Li-Fraumeni Mouse Models". We would be happy to publish your paper in Life Science Alliance pending final revisions necessary to meet the reviewers' requests and our formatting guidelines.

As you will note from the comments below, the reviewers are quite excited about these findings and have only requested minor text edits to the study. Thus, we would like to invite you to submit a final revision addressing these minor concerns and our formatting requests.

Along with the points listed below, please also attend to the following:

- please upload your main manuscript text as an editable doc file
- please upload your figures as single files
- please add ORCID ID for Corresponding Author
- please add a Category and Summary blurb/Alternate Abstract in our system
- please add a conflict of interest statement to your main manuscript text
- please use the [10 author names, et al.] format in your references (i.e. limit the author names to the first 10)
- please upload your Tables as separate editable excel or doc files

A. FINAL FILES:

B. MANUSCRIPT ORGANIZATION AND FORMATTING:

Sincerely,

Shachi Bhatt, Ph.D.
Executive Editor
Life Science Alliance

Reviewer #1 (Comments to the Authors (Required)):

This manuscript by Chan et al describes an interesting genetic study to determine possible genetic and stochastic effects on tumor formation and the type of tumors that might be due to inherited mutant p53s often identified in Li-Fraumeni syndrome (LFS) patients by employing a genetically manipulated mouse model system that harbor Li-Fraumeni syndrome-derived p53 mutants. Their results demonstrate that p53 mutants have a strong impact on tumorigenesis and the type of tumors that are formed during this process, which somewhat mimics the LFS phenotype. This genetic study is interesting and important. Addressing the following minor points might help them improve their manuscript.

1. In the Abstract, the sentence "The genetic background of mice carrying Tp53 mutations has a strong influence upon the tissue type of the tumor produced as well as the formation of multiple tumors produced in a single mouse" repeats twice. Can this claim be more concisely expressed?
2. In Table 1, the most common tumors in mice are T-cell lymphomas, which are not found in humans with Li-Fraumeni syndrome. Is this because the mice are prone to grow lymphomas independent of p53 status? An explanation would be helpful.
3. Some opinions or points in this manuscript may need their related references: 1) Introduction: "Background also makes ... are predisposed to breast cancers"; 2) Result-Figure 1: "Based upon previous observations, this was done most intensively between 11 and 20 months after birth." Please either cite the result or a reference(s); 3) Result-Figure 4: To support the reason that the experiment was stopped at 20 months after birth.
4. Some proofs would be helpful to improve their presentation (such as pictures of H&E staining slides or tissues) of anatomical observation for Figures 1, 2 and 4).
5. In Fig 1, please explain when mice were euthanized, the mice were randomly chosen?
6. In Fig 3, why were the two strains, A/J and NOD, chosen?

Reviewer #2 (Comments to the Authors (Required)):

Chan et al
Genetic and Stochastic influences upon tumor formation...

In this manuscript, Hu and colleagues cross R172H mice in a C57Bl/6 background to 7 other genetic backgrounds and they analyze the incidence and timing of cancer in the different F1 progeny. They report some fascinating differences: SWR and NOD strains develop cancer the latest, but SWR also has the highest incidence of cancer. F1 progeny crossed into A/J and NOD backgrounds have dramatically different median survivals, and Balb/c mice have a phenotype most closely resembling, to this reviewer, to Li Fraumeni in humans (high rates of sarcoma, plus quite a bit of carcinoma, as

well as high numbers of mice with multiple tumors). The authors conclude that the existence of dominant genetic modifiers in these different strains is likely at play; they also conservatively and correctly conclude that other differences, such as microbiome, gender/sexual dimorphism and epigenetics may also be factors in these differences. In all this manuscript is a very useful resource for individuals using GEMM models to assess the impact of p53 mutation on cancer. Two very minor changes are requested:

1. Figure 1 and 2 differ in that they are based on observational versus histopathological data, but the figures are related; I would like the authors to use the same colors and also to label each line, as done in Figure 1, in Figure
2. Page 11 "This is the reason..." paragraph should be attached to the preceding paragraph (starts with C57Bl/6 mice)
3. Table 1: re-label "Incidence of tumor types" to be "Average incidence of tumor types"

Following is our response to the comments from reviewers.

Reviewer #1 (Comments to the Authors (Required)):

This manuscript by Chan et al describes an interesting genetic study to determine possible genetic and stochastic effects on tumor formation and the type of tumors that might be due to inherited mutant p53s often identified in Li-Fraumeni syndrome (LFS) patients by employing a genetically manipulated mouse model system that harbor Li-Fraumeni syndrome-derived p53 mutants. Their results demonstrate that p53 mutants have a strong impact on tumorigenesis and the type of tumors that are formed during this process, which somewhat mimics the LFS phenotype. This genetic study is interesting and important. Addressing the following minor points might help them improve their manuscript.

1. In the Abstract, the sentence "The genetic background of mice carrying Tp53 mutations has a strong influence upon the tissue type of the tumor produced as well as the formation of multiple tumors produced in a single mouse" repeats twice. Can this claim be more concisely expressed?

Response: Thank you for this good suggestion. We revised the sentence as following:
The genetic background of mice carrying Tp53 mutations has a strong influence upon the tissue type of the tumor produced as well as the number of tumors formed in a single mouse.

2. In Table 1, the most common tumors in mice are T-cell lymphomas, which are not found in humans with Li-Fraumeni syndrome. Is this because the mice are prone to grow lymphomas independent of p53 status? An explanation would be helpful.

Response: This is a very good suggestion. It has been observed that some of the inbred strains, such as Balb/c and C57BL/6, are predisposed to developing certain spontaneous tumors, with lymphoma often being most common, indicating that the mouse and human T cell adaptive immune systems have a differential dependence upon p53 to prevent a lymphoma or leukemia (Kohnken R., et al. Front Oncol. 2017, 7:22). We have added this explanation in Page 7, 2nd paragraph, line 4-8.

3. Some opinions or points in this manuscript may need their related references: 1) Introduction: "Background also makes ... are predisposed to breast cancers"; 2) Result-Figure 1: "Based upon

previous observations, this was done most intensively between 11 and 20 months after birth." Please either cite the result or a reference(s); 3) Result-Figure 4: To support the reason that the experiment was stopped at 20 months after birth.

Response: As the reviewer suggested, we have added references in all above-mentioned places to support those statements.

4. Some proofs would be helpful to improve their presentation (such as pictures of H&E staining slides or tissues) of anatomical observation for Figures 1, 2 and 4).

Response: This is a very good suggestion. As suggested, a panel of representative H&E images of tumors, including Thymic lymphoma, adenocarcinoma, sarcoma, osteosarcoma, liposarcoma and angiosarcoma, are presented in Figure 2B.

5. In Fig 1, please explain when mice were euthanized, the mice were randomly chosen?

Response: Mice were randomly chosen at the time of euthanization for results presented in Figures 1. We have clarified this in the manuscript text.

6. In Fig 3, why were the two strains, A/J and NOD, chosen?

Response: A/J F-1 and NOD F-1 hybrid mice were chosen because these 2 strains showed the biggest difference in the onset of tumor formation as shown in Figure 1; AJ was the fastest strain to develop tumors in p53 heterozygous mice while NOD is the slowest strain. We have added this explanation in Page 6 last line to Page 7 line 3.

Reviewer #2 (Comments to the Authors (Required)):

In this manuscript, Hu and colleagues cross R172H mice in a C57Bl/6 background to 7 other genetic backgrounds and they analyze the incidence and timing of cancer in the different F1 progeny. They report some fascinating differences: SWR and NOD strains develop cancer the latest, but SWR also has the highest incidence of cancer. F1 progeny crossed into A/J and NOD backgrounds have dramatically different median survivals, and Balb/c mice have a phenotype most closely resembling, to this reviewer, to Li Fraumeni in humans (high rates of sarcoma, plus quite a bit of carcinoma, as well as high numbers of mice with multiple tumors). The authors conclude that the existence of dominant genetic modifiers in these different strains is likely at play; they also conservatively and correctly conclude that other differences, such as microbiome, gender/sexual dimorphism and epigenetics may also be factors in these differences. In all this manuscript is a very useful resource for individuals using GEMM models to assess the impact of p53 mutation on cancer. Two very minor changes are requested:

1. Figure 1 and 2 differ in that they are based on observational versus histopathological data, but the figures are related; I would like the authors to use the same colors and also to label each line, as done in Figure 1

Response: This is a very good suggestion. We have revised Figure 2 so that both figures (Figure 1 and 2) use the same color for each F-1 strain).

2. Page 11 "This is the reason..." paragraph should be attached to the preceding paragraph (starts with C57Bl/6 mice)

Response: Thanks for this good suggestion. We have regrouped 2 paragraphs as suggested.

3. Table 1: re-label "Incidence of tumor types" to be "Average incidence of tumor types"

Response: We have re-labelled Table 1 as suggested.

In addition to addition responses and changes, as requested in the decision letter, we have formatted the manuscript according to your journal's guideline. We have provided Summary blurb and added a conflict of interest statement in the manuscript test.

We hope that with these changes and responses, the manuscript is acceptable for publication. Again, we really appreciate the great efforts and the insightful and constructive comments of you and expert reviewers.

Thank you very much for your kind consideration.

Sincerely yours,

Wenwei Hu, Ph.D.
Associate Professor
Rutgers Cancer Institute of New Jersey
New Brunswick, NJ 08903
Email: wh221@cinj.rutgers.edu
Phone: 732-235-6169

December 15, 2020

RE: Life Science Alliance Manuscript #LSA-2020-00952-TR

Dr. Wenwei Hu
Rutgers Cancer Institute of New Jersey
Radiation Oncology
195 Little Albany Street
New Brunswick 8903

Dear Dr. Hu,

Thank you for submitting your Research Article entitled "Genetic and Stochastic Influences upon Tumor Formation and Tumor Types in Li-Fraumeni Mouse Models". It is a pleasure to let you know that your manuscript is now accepted for publication in Life Science Alliance. Congratulations on this interesting work.

*** We are still waiting for the 2nd corresponding author to update their ORCID ID in the system. We will only be able to proceed further with the ORCID ID in hand. ***

DISTRIBUTION OF MATERIALS:

Again, congratulations on a very nice paper. I hope you found the review process to be constructive

and are pleased with how the manuscript was handled editorially. We look forward to future exciting submissions from your lab.

Sincerely,

Shachi Bhatt, Ph.D.

Executive Editor

Life Science Alliance

<https://www.lsjournal.org/>
